# Enhancing Health Literacy and Self-Management in Glaucoma Patients: Evidence from a Nurse-Led Educational Intervention

**DOI:** 10.3390/healthcare13080861

**Published:** 2025-04-09

**Authors:** Lοukia Tsichla, Evridiki Patelarou, Efstathios Detorakis, Miltiadis K. Tsilimbaris, Athina E. Patelarou, Konstantinos Giakoumidakis

**Affiliations:** 1Department of Nursing, School of Health Sciences, Hellenic Mediterranean University, 71410 Heraklion, Crete, Greece; ddk192@edu.hmu.gr (L.T.); epatelarou@hmu.gr (E.P.); apatelarou@hmu.gr (A.E.P.); 2School of Medicine, University of Crete, Voutes, 71110 Heraklion, Crete, Greece; detorakis@hotmail.com (E.D.); tsilimb@gmail.com (M.K.T.)

**Keywords:** glaucoma, health literacy, self-management, nurse-led intervention, patient education, chronic disease

## Abstract

**Objective:** This interventional cohort study evaluates the effectiveness of a nurse-led educational intervention designed to improve health literacy and self-management in glaucoma patients and examines correlations with demographic factors (sex, age, education level) and comorbidities. **Material and Methods:** A convenience sample of 312 glaucoma patients was recruited from the University General Hospital of Heraklion, Crete, between November 2022 and November 2023. The patients were randomly assigned to an intervention group (receiving nurse-led education) or a control group (receiving standard care). Health literacy was measured using the HLS-EU-16 questionnaire, while self-management levels were assessed with the Patient Activation Measure (PAM-13). Two-way repeated measures ANOVA analyzed intervention effects over time. Moreover, multiple linear regression analysis was used to test for potential correlations between variables. **Results:** Significant improvements were observed in both HLS-EU-16 and PAM-13 scores in the intervention group over time compared to the control group. In the post-test, the intervention group showed statistically higher health literacy scores (12.79 ± 2.64) than the control group (10.52 ± 3.60), with a mean difference in −2.27 (*p* < 0.001). Similarly, the PAM-13 scores were significantly higher in the intervention group (49.35 ± 10.36) compared to the control group (41.33 ± 24.12) post-test, with a mean difference of −8.01 (*p* < 0.001). The pre-intervention PAM-13 score was positively associated with both the HLS-EU16 score (B = 3.307, *p* < 0.001) and education level (B = 3.863, *p* = 0.037). Meanwhile, higher post-intervention PAM-13 was positively associated with greater post-intervention HLS-EU16 scores (B = 1.26, 95% CI [0.61, 1.91]). Furthermore, the participants of the intervention group exhibited significantly higher post-intervention PAM-13 scores than the control group (B = 5.36, 95% CI [1.03, 9.68]). **Conclusions:** The nurse-led educational intervention significantly enhanced health literacy and self-management in glaucoma patients, indicating its potential value in patient education strategies for chronic diseases.

## 1. Introduction

Glaucoma is a leading cause of irreversible blindness globally, characterized by the progressive degeneration of the optic nerve and the loss of the visual field [1]. The disease is progressive and asymptomatic in its early stages, making many patients unaware of the early signs [1]. Timely diagnosis and effective management are crucial for preserving vision [2]. Treatment approaches include long-term adherence to pharmacological regimens, regular monitoring, and, in some cases, surgical interventions [3]. More importantly, the successful management of glaucoma is often influenced by factors such as health literacy and patients’ self-management abilities [4]. However, a critical barrier to successful management is the lack of adequate health literacy among many glaucoma patients [4]. While the complexity of the disease necessitates a strong understanding of its nature and the ability to manage treatment plans effectively, a significant proportion of patients struggle to comprehend their condition and the importance of adherence to prescribed medication [5]. This gap in health literacy often leads to inconsistent treatment, missed follow-ups, and suboptimal outcomes, further exacerbating the disease’s impact on patients. Addressing this deficiency is crucial, as health literacy serves as the foundation for informed decision-making and sustained treatment adherence [5].

Therapeutic adherence is a pivotal determinant of clinical outcomes in glaucoma management. Given the chronic nature of the disease, lifelong adherence to complex treatment regimens is essential. However, a study has indicated that 30–80% of patients exhibit suboptimal adherence to topical therapy [6]. Recognizing this challenge, our study was specifically designed to address adherence-related barriers and assess the impact of the proposed intervention on patient compliance

Self-management is a cornerstone of successful management of glaucoma and other chronic diseases [3]. It encompasses the patient’s active healthcare participation, involving appropriate medication utilization, disease progression monitoring, and informed treatment decisions [4]. Patients with developed self-management skills exhibit improved medication adherence, contributing to disease stabilization and enhanced clinical outcomes [5]. Conversely, limited self-management abilities are associated with poor medication adherence, delayed healthcare visits, and accelerated disease progression [5]. Effective self-management necessitates adequate education and support, which, in many instances, falls short [7]. Health literacy, on the other hand, holds equal importance for glaucoma management. It refers to the patient’s capacity to acquire, comprehend, and utilize health-related information to make informed decisions about their care [8]. Studies have demonstrated that patients with low health literacy encounter challenges in comprehending the necessity of regular visits, the long-term nature of treatment, and the significance of proper medication adherence [8]. Consequently, a lack of health literacy often leads to delayed medical consultations, improper medication usage, and inadequate understanding of disease progression [9]. This context underscores the imperative for educational interventions aimed at enhancing both health literacy and self-management skills. Towards that end, various educational programs for glaucoma patients have been developed [10]. However, most interventions primarily focus on theoretical knowledge of the disease, neglecting practical aspects of self-management and tailoring content to individual patient needs [10,11]. In particular, nurse-led educational interventions represent a promising solution to address these gaps.

Nurse-led interventions could potentially improve the health literacy and self-management of patients with glaucoma due to their holistic, accessible, and patient-centered nature. These interventions are not only cost-effective but also contribute to improved clinical outcomes by empowering patients with greater health literacy and self-management skills [12]. It is because nurses already build strong, trusting relationships through frequent interactions, enabling them to address clinical, psychological, social, and lifestyle factors that affect health [13]. Their consistent involvement allows them to monitor progress, adjust care plans, and explain medical information clearly [14]. In addition, the individualized holistic approach that nurses usually follow makes them ideal for patient education, as they can build trust and promote sustained behavior change in patients [15]. However, gaps remain in understanding how tailored nurse-led interventions impact specific chronic conditions like glaucoma, where health literacy and self-management are critical to preventing or delaying severe visual impairment [16].

In Greece, both health literacy and patient activation levels remain notably low [17], posing significant challenges in managing chronic conditions like glaucoma [18]. Moreover, there is a lack of structured education and support programs for individuals with chronic diseases which has resulted in lower adherence to treatment protocols and worsened healthcare outcomes [19]. A study that would provide an intervention to improve health literacy and patient self-management activation level could provide valuable insights into optimizing educational interventions in glaucoma management [4]. Therefore, the present study aimed to evaluate the effect of a nurse-led educational intervention tailored to patients with glaucoma, with the goal of introducing new insights to the existing body of knowledge.

## 2. Material and Methods

### 2.1. Study Design

An interventional cohort study was conducted on 312 patients with glaucoma at the outpatient clinic of a university hospital in Heraklion, Crete, Greece. The population of the present study consisted of patients diagnosed with glaucoma at least six months before the study. Notably, the diagnosis of glaucoma was based on the following diagnostic criteria: (a) high intraocular pressure, intraocular pressure (IOP) > 21 mmHg measured via Goldmann applanation tonometry; (b) quality control measures (dual verification); (c) optic nerve atrophy assessed through fundoscopic examination and optical coherence tomography (OCT); and (d) deficits in the visual fields [20,21]. Moreover, the glaucoma of these patients was not in an early stage, as evidenced by extensive visual field defects (MD > −8.5 dB) [3]. All diagnostic data were cross-validated against medical records by two independent ophthalmologists to ensure accuracy.

A convenience sample of glaucoma involved patients who attended the outpatient department of the Ophthalmology clinic of the University General Hospital of Heraklion (Crete, Greece) from November 2023 to May 2024. The study inclusion criteria were as follows: (i) age ≥ 18 years; (ii) informed written consent to participate in the study; (iii) adequate knowledge of the Greek language (writing and reading); (iv) diagnosis of the condition for at least six months; and (v) “stable medical therapy for glaucoma over the past three months” to ensure that participants were under consistent treatment, minimizing variability that could influence the study outcomes. Patients with incomplete information recorded in their health records or patients who did not fully complete each questionnaire were excluded (n = 37). After signing the informed consent, the study participants were allocated in random chronological order into two groups. These were, namely, (a) the control group (patients with an odd registration number) and (b) the intervention group (patients with an even registration number) (Figure 1). Participants were allocated in a 1:1 ratio using sequential registration number assignment (even/odd), ensuring balanced groups (156 patients per arm). Baseline comparisons confirmed no significant differences in key characteristics (age, sex, education, comorbidities), supporting sample homogeneity (Table 1).

### 2.2. Description of This Study’s Intervention

The intervention group received our nurse-led intervention, which included (a) a 15 min oral training session by the first author, (b) provision of printed material (brochure), and (c) provision of appropriately selected online videos from the World Glaucoma Association (Table 2). During the oral training session, the researcher explained the key aspects of the disease in detail. Specifically, the identification and understanding of symptoms helped patients recognize early warning signs and monitor their condition. The training also guided medication use, explaining how and when prescribed medications should be taken, possible side effects, and the importance of adhering to the treatment plan. Before ending the session, patients were instructed to watch selected videos from the World Glaucoma Association which offer key information about glaucoma, aiming to raise awareness among the public and patients. These videos were selected since they covered important aspects of glaucoma management, such as the nature of glaucoma, its symptoms, and the critical importance of early detection. The videos also discuss glaucoma causes, risk factors, and available treatment options, helping viewers understand how to manage their disease more effectively. Finally, the participants were also provided with a brochure that contained pertinent information regarding glaucoma, treatment options, and self-care strategies.

The holistic approach of the program aimed to empower patients with the knowledge and skills necessary for long-term disease management and overall well-being. The oral training session took place in an appropriate, quiet, and clean space, respecting the patients’ privacy. During this process, the study participants had the opportunity and time to ask questions and request clarifications or additional information.

### 2.3. Data Collection and Instruments

Data collection for both groups regarding the level of health literacy and patient activation was carried out by the research team through an in-person interview using three questionnaires and an anonymous self-report questionnaire (developed for this study), along with the Greek versions of the European Health Literacy Survey Questionnaire 16 (HLS-EU-16), and the Patient Activation Measure-13 (PAM-13). The self-report questionnaire was used to collect the demographic characteristics of participants, including (a) biological sex, (b) age, (c) educational level (primary, secondary, tertiary), and (d) comorbidity (no-yes). These assessments were conducted at two pivotal time points: a pre-test (before the intervention) and a post-test (following the intervention). The pre-test served to establish baseline levels of health literacy and self-management for both the control and intervention groups. Conversely, the post-test enabled the assessment of changes that transpired after the intervention. The intervention was based on the Health Belief Model (HBM) and the Self-Management Support Framework. The HBM guided the focus on patient education regarding glaucoma severity, symptoms, and treatment benefits to enhance understanding and adherence [22]. The Self-Management Support Framework provided the foundation for empowering patients with the skills and resources necessary for effective self-care and long-term management. This combined approach aimed to improve both health literacy and self-management in glaucoma patients [23].

#### 2.3.1. European Health Literacy Survey Questionnaire 16 (HLS-EU-16)

The European Health Literacy Survey Questionnaire 16 (HLS-EU-16) was employed to assess participants’ health literacy. This instrument has been translated into Greek and validated for use within the Greek population, as documented in prior research [24]. Greek version of the European Health Literacy Survey Questionnaire 16 (HLS-EU-16) was used [24]. The questionnaire is available in three versions depending on the number of questions (47, 16 or 6). Its short form, which was developed based on the Rasch model, distinguishes three levels of literacy: adequate health literacy, problematic health literacy, and insufficient health literacy. It includes 16 questions, each of which is answered on a 4-point Likert scale ranging from 1 (“very easy”) to 4 (“very difficult”). Responses of “very easy” and “easy” receive a point, while “very difficult” and “difficult” do not receive a point. To calculate the total score of the questionnaire, the score of each answer is added, with the final score having a range of 0–16. Scores between 0 and 8 indicate inadequate health literacy, scores between 9 and 12 indicate problematic health literacy, and scores between 13 and16 indicate sufficient health literacy [25].

#### 2.3.2. The Patient Activation Measure-13 (PAM-13)

The Patient Activation Measure-13 (PAM-13) questionnaire was used to assess the patient’s level of activity. It was created by Hibbard JH [15] as the short version of the original 22-item questionnaire [1]. Additionally, this tool has been translated into the Greek language and validated within the Greek population [26]. It has been used primarily in patient populations with chronic diseases, but also at the level of primary healthcare. More specifically, it has been evaluated in a series of patients with chronic diseases, elderly patients with commodities, patients with surgical health problems, and patients suffering from neurological diseases, diabetes mellitus, and osteoarthritis. The PAM-13 includes 13 questions, each of which is answered on a 4-point Likert scale ranging from 1 (“strongly disagree”) to 4 (“strongly agree”), with the additional option “not applicable”. To calculate the total score of the questionnaire, the score of each answer is summed and the sum is divided by the number of answers given by the study participant, excluding the “not applicable” answers. Then, this resulting score is algebraically transformed into the final score, having a range of 0–100. Higher PAM score values are indicative of higher patient activity levels [27,28]. Patients are classified into four levels of activity: (a) Level 1 with a score of ≤47.0 (disengaged and overwhelmed), (b) Level 2 with a score of 47.1–55.1 (becoming aware but still struggling), (c) Level 3 with a score of 55.2–67.0 (taking action and gaining control), and (d) Level 4 with a score of ≥67.1 (maintaining behavior and pushing further) [27].

### 2.4. Statistical Analysis

We performed the statistical analysis using SPSS version 26.0. Continuous variables were expressed as mean ± standard deviation and categorical variables were expressed by frequency and percentage. Internal consistency was assessed by Cronbach’s alpha. A Cronbach α coefficient >0.7 indicates acceptable reliability for research purposes and suggests that items are interdependent and homogeneous in terms of the construct they measure. The two-way repeated measures ANOVA is used to determine whether any change in HLS-EU-16 or PAM-13 scores (the dependent variables) is the result of the interaction between the use of nurse-led training (which is one of our factors) and “time” (our second factor). To control the confounding of demographics, when studying the correlation between the HLS-EU-16 (independent variable) and PAM-13 scale (dependent variable) in pre- and post-intervention tests, multiple linear regression analysis was used. The independent variables included group, biological sex, age, education level, comorbidity, and HLS-EU16 score. Assumptions for linear regression were checked and fulfilled. For all tests, statistical differences were determined to be significant at *p* < 0.05.

## 3. Results

The study sample consisted of 312 patients, 134 women (42.9%) and 178 men (57.1%), with a mean age of 63.9 ± 14.4 years. Among them, 228 patients (73.1%) had primary, 44 (14.1%) secondary, and 40 (12.8%) tertiary educational levels. Two hundred eighty-seven (92%) patients had comorbidities, including hypertension, diabetes, cardiovascular diseases, and hyperlipidemia. Cronbach’s alphas for HLS-EU16 and PAM-13 were 0.88 and 0.95, respectively. There was not a statistically significant difference between groups: (a) age [t(310) = −0.42, mean difference −0.68 (95% CI, −3.89 to 2.52), *p* = 0.674], (b) biological sex [x2(1) = 1.027, *p* = 0.360], (c) comorbidity [x2(1) = 0.031, *p* = 0.861], and (d) educational level [x2(2) = 1.968, *p* = 0.374] (Table 1).

The analysis examines the effect of different interventions over time on HLS-EU-16 and PAM-13 scores.

There was a statistically significant interaction between intervention and time on the HLS-EU-16 score, F (1, 310) = 49.84, *p* < 0.001. Therefore, simple main effects were run. HLS-EU-16 score was not statistically significantly different in the control group (10.34 ± 3.80) compared to the intervention group (11.12 ± 3.54) at the pre-test, t (310) = −1.87, *p* = 0.062, a mean difference of −0.78 (95% CI, −1.60 to 0.04). However, the HLS-EU-16 score was statistically significantly different in the control group (10.52 ± 3.60) compared to the intervention group (12.79 ± 2.64) at the post-test, with t (310) = −6.33, *p* < 0.001, and a mean difference of −2.27 (95% CI, −2.97 to −1.56) (Figure 2).

There was a statistically significant interaction between intervention and time on PAM-13 score, with F (1, 310) = 19.74 and *p* < 0.001. Therefore, simple main effects were run (Table 3). The PAM-13 score was not statistically significantly different in the control group (41.4 ± 24.2) compared to the intervention group (40.1 ± 23.7) at the pre-test, with t (310) = 0.50, *p* = 0.615, and a mean difference of 1.37 (95% CI, −3.97 to 6.70). However, the PAM-13 score was statistically significantly different in the control group (41.3 ± 24.1) compared to the intervention group (49.3 ± 10.4) at the post-test, with t (310) = −3.81, *p* < 0.001, and a mean difference of −8.01 (95% CI, −12.16 to −3.87) (Figure 3).

A multiple regression analysis was conducted to predict pre-intervention PAM-13 scores based on HLS-EU16 score, group assignment, biological sex, age, education level, and comorbidity status. The overall model was statistically significant, with F (6, 305) = 20.079 and *p* < 0.001, explaining 27% of the variance in PAM-13 scores (adj. R^2^ = 0.27). Significant positive predictors of the pre-intervention PAM-13 score were the HLS-EU16 score (*p* < 0.001) and education level (*p* = 0.037). Specifically, higher pre-intervention PAM-13 values were independently associated with greater pre-intervention HLS-EU16 scores (B = 3.31, 95% CI [2.68, 3.93]) and with higher education level (B = 3.86, 95% CI [0.24, 7.49]) (Table 4).

Furthermore, the multiple regression model was a statistically significant predictor of the post-intervention PAM-13 score, with F (6, 305) = 5.627 and *p* < 0.001, and with an adjusted R^2^ of 0.082. Both the HLS-EU16 (*p* < 0.001) and group (*p* = 0.015) scores contributed significantly to the prediction (Table 5). Specifically, higher post-intervention PAM-13 values were independently associated with greater post-intervention HLS-EU16 scores (B = 1.26, 95% CI [0.61, 1.91]) and with group allocation, as participants in the intervention group exhibited significantly higher post-intervention PAM-13 scores than the control group (B = 5.36, 95% CI [1.03, 9.68]).

## 4. Discussion

The present study aimed to assess the effect of a nurse-led educational intervention on health literacy and self-management among patients with glaucoma attending outpatient clinics at a tertiary hospital in Greece. Our findings suggest that our intervention has improved health literacy and self-management of patients with glaucoma. These findings also suggest that structured, nurse-led educational intervention may enhance glaucoma patients’ ability to better understand and manage their condition.

A major finding of the present study was the substantial improvement in health literacy among glaucoma patients who participated in the nurse-led educational intervention. While our randomization approach (based on registration numbers) accommodated clinical workflow constraints, we acknowledge that computer-generated randomization would strengthen future replications of this intervention. These findings are in line with previous research, such as that of Silva et al. [29], who reported that structured health literacy interventions lead to a better understanding of chronic diseases and improved adherence to treatment plans. Additionally, Babcock et al. [30] emphasized that interactive and patient-specific educational strategies sustain health literacy improvements over time, further supporting the efficacy of the current intervention. Moreover, the World Health Organization has also highlighted the critical role of trust in health literacy interventions, where consistent interactions with healthcare professionals were associated with better patient outcomes [29]. A potential explanation for the improvement in health literacy in our study could be due to the personalized and interactive nature of the educational intervention. Unlike more traditional, didactic methods, our intervention utilized tailored content and active engagement strategies, allowing participants to address their unique informational needs and health-related challenges.

Cultural factors play a critical role in shaping health literacy and self-management behaviors. In Greece, strong family involvement in healthcare decisions may influence patient adherence and educational interventions [31]. Additionally, socioeconomic factors, cultural values, and social networks play significant roles in health-related behaviors. Future adaptations of this intervention should consider regional variations in health beliefs, physician–patient dynamics, and accessibility of health information to optimize effectiveness [32].

The study also revealed a significant improvement in self-management and activation capabilities among participants in the intervention group. This improvement in self-management is critical for the long-term management of chronic diseases and aligns with the existing literature. For instance, Bodenheimer et al. (2002) [33] found that chronic disease self-management programs (CDSMP) delivered by healthcare professionals enhance patient confidence and adherence to care plans. Similarly, other studies [33] have shown that nurse-led programs significantly foster self-efficacy and informed decision-making, ultimately reducing healthcare utilization. Moreover, a systematic review also indicates that self-management support interventions could improve adherence and health behaviors in chronic diseases like diabetes and cardiovascular conditions [34]. In addition, Nutbeam et al. (2008) suggested that interventions targeting health literacy and self-management not only improve clinical outcomes but also empower patients to take a proactive role in their care [35]. Moreover, systematic reviews [36] have consistently highlighted the importance of multidisciplinary approaches, including nurse-led and interprofessional education, in optimizing patient activation and adherence Additionally, studies on chronic kidney disease have demonstrated that activation-focused interventions significantly enhance self-management behaviors, dietary compliance, and medication adherence [37]. This is important, since higher levels of activation, reflecting greater confidence and skills in managing health, are strongly associated with better health behaviors and outcomes [1,37]. This is especially true for glaucoma patients, as they must adhere to lifelong, complex treatment regimens, such as regular use of eye drops and frequent ophthalmic evaluations.

The findings of the present study emphasize the importance of patient education in the care of glaucoma patients, confirming previous research. For example, a study by Sleath et al. (2015) [16], which analyzed the role of providers in educating patients about glaucoma and its pharmacological treatment also highlighted the importance of patient education. Similarly, Tsai (2009) [38] provided a comprehensive perspective on patient adherence to topical glaucoma therapy, emphasizing the role of education in improving self-management. Additionally, the study by Briesen et al. (2010) [39] on the reasons for refusal of cataract surgery underscores the significance of education in facilitating acceptance and adherence to therapeutic interventions—an insight that can also be applied to glaucoma management. Therefore, strengthening educational interventions is a key factor in improving the self-management of glaucoma patients, ultimately leading to better therapeutic outcomes.

One of the most significant findings of this study was the positive association between health literacy and patient activation, following our nurse-led educational intervention. These findings align with the existing literature, which emphasizes health literacy as a key determinant of patient self-management, treatment adherence, and effective healthcare system navigation [29,40]. Furthermore, another study highlights that low health literacy can exacerbate healthcare disparities by limiting patients’ ability to understand medical instructions and implement self-management strategies [41,42]. On the other hand, studies have shown that structured intervention programs, similar to our intervention, that simplify medical information could promote health literacy skills, and encourage active patient participation, leading to reduced hospital readmissions, improved medication adherence, and enhanced overall patient satisfaction [35,43]. Additionally, the integration of health literacy assessments into routine clinical care could help healthcare providers identify at-risk patients and tailor interventions to meet their specific needs, thereby addressing disparities in healthcare access and outcomes [44]. These approaches also align with the World Health Organization’s call for health systems to adopt patient-centered strategies that prioritize health literacy as a means of achieving universal health coverage and reducing inequities [45]. Therefore, by fostering a healthcare environment that values and promotes health literacy and patient activation, policymakers and healthcare practitioners could empower patients to take an active role in managing their health, ultimately leading to more sustainable and equitable healthcare systems. Future studies could concentrate on target areas to build upon the findings of this study. Specifically, the findings related to the impact of patient activation on glaucoma management and treatment adherence should be explored further. In this regard, educational programs should be designed to assess how patient activation can enhance the management of glaucoma and improve adherence to prescribed treatments. These programs must incorporate tools to promote self-management, such as mobile applications and telemedicine sessions, ensuring that education reaches patients remotely, particularly those in underserved areas. For glaucoma patients, educational programs should be structured to demystify the condition, utilizing interactive tools that help patients understand glaucoma, its progression, and the importance of regular eye check-ups. Practical training on the proper use of medications, such as workshops for administering eye drops correctly, should be a core component. Furthermore, the development of telemedicine platforms could provide ongoing support by facilitating health monitoring, regular check-ins, and feedback for the patients. On the other hand, studies have shown that adherence monitoring techniques significantly improve patient compliance by providing objective data on medication use [46] and intraocular pressure (IOP) fluctuation indices, highlighting the critical role of these methods in glaucoma management. Additionally, IOP variability has been identified as an independent risk factor for disease progression, as large fluctuations in IOP are associated with faster disease advancement [47,48]. These techniques enable nurses to tailor patient education with precision, enhancing adherence and improving clinical outcomes. Furthermore, the implementation of structured educational protocols has been shown to increase patient adherence by 40% and reduce the rate of retinal nerve fiber layer (RNFL) loss by 15% over two years [49,50]. These findings underscore the transformative potential of nurse-led interventions in preserving vision and preventing disease progression, making their integration into daily clinical practice not just an improvement but a necessity for modern ophthalmic care.

### Limitations

This study presents an innovative approach to glaucoma management, employing nurse-led educational interventions to enhance patient activation and health literacy. While these promising outcomes warrant acknowledgment, the study also faces several limitations that should be considered. Firstly, the study’s focus on a single geographic region, Crete, may restrict the generalizability of the findings. The sample reflects typical glaucoma patients in a tertiary referral center, though findings may not fully generalize to broader populations. Future research should prioritize multicenter validation across diverse healthcare settings (e.g., primary care, urban and rural hospitals) to assess the intervention’s applicability. Although the insights gained are valuable, a more diverse and representative sample from multiple centers would ensure that the conclusions apply to a broader range of patients with glaucoma. Secondly, the relatively short duration of the study/intervention precludes our ability to assess the long-term impact of the intervention. Therefore, extended follow-up assessments are suggested to provide a clearer picture of whether the improvements in health literacy and patient activation persist and lead to sustained behavioral changes and health outcomes. Thirdly, while the intervention was comprehensive in its delivery methods, it was conducted as a single session without ongoing support. This may have limited the retention and application of the information provided. Fourthly, all outcome measures in this study were self-reported, which may introduce recall and social desirability biases. While validated instruments were used, the absence of objective adherence metrics (e.g., pharmacy refill rates, electronic monitoring) remains a limitation. Future studies should integrate objective adherence measures, such as pharmacy refill data or electronic medication monitoring, to enhance the validity of findings and provide a more comprehensive evaluation of patient adherence patterns. Fifthly, the positive results of the intervention should be interpreted with caution, since these observed improvements may also reflect the high engagement level of the intervention group, potentially influenced by its structured format, the interaction with the research team, and potential biases related to self-reported measures. Future studies could evaluate the effectiveness of periodic follow-up sessions, potentially utilizing digital platforms, to sustain learning and patient engagement over time. Furthermore, testing the nurse-led intervention in different chronic disease populations could provide valuable insights into its applicability and effectiveness. Finally, one notable limitation of this study is the use of convenience sampling, which may have limited the generalizability of the findings to broader populations. Future research should consider employing a formal randomization process to enhance methodological robustness and ensure more representative sampling.

## 5. Conclusions

In conclusion, the findings of the present study have highlighted that a nurse-led educational intervention could enhance glaucoma management by improving both health literacy and self-management among outpatient glaucoma patients. Another major finding of the present study was that health literacy and self-management were positively associated. Healthcare managers, educators, and policymakers could utilize these findings to develop and implement comprehensive educational programs that aim to improve patient outcomes in glaucoma care. By implementing these educational interventions, healthcare systems can transition towards a more proactive and preventive approach to care, thereby reducing the long-term burden associated with diseases such as glaucoma and enhancing the overall quality of life for patients. Future studies can further leverage these findings to develop similar interventions that improve the management of glaucoma and other chronic diseases.

## Figures and Tables

**Figure 1 healthcare-13-00861-f001:**
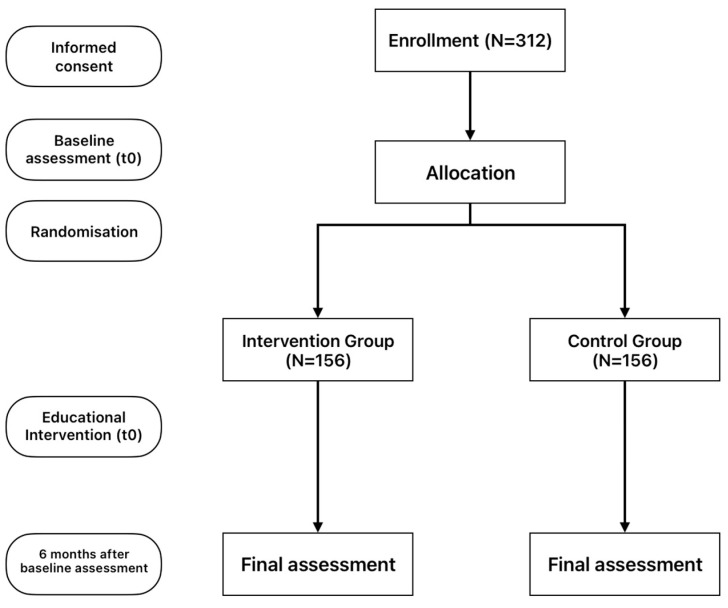
Study flowchart depicting participant allocation and assessment timeline.

**Figure 2 healthcare-13-00861-f002:**
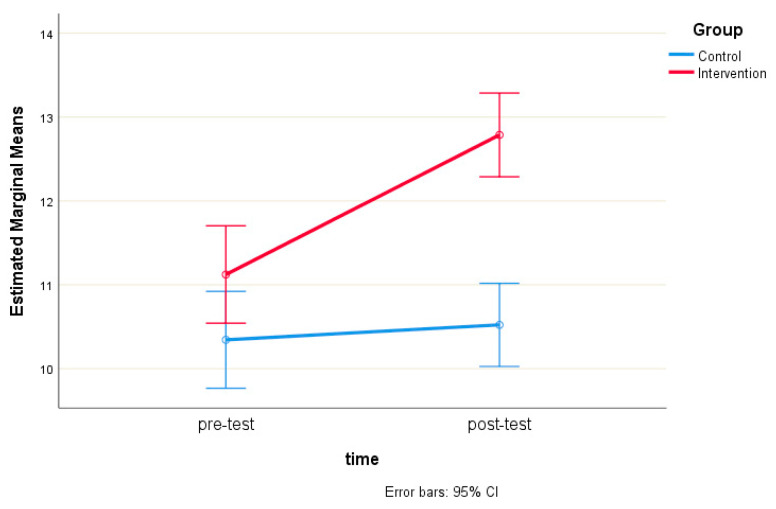
Estimated marginal means of HLS-EU-16 scores in the intervention and control groups.

**Figure 3 healthcare-13-00861-f003:**
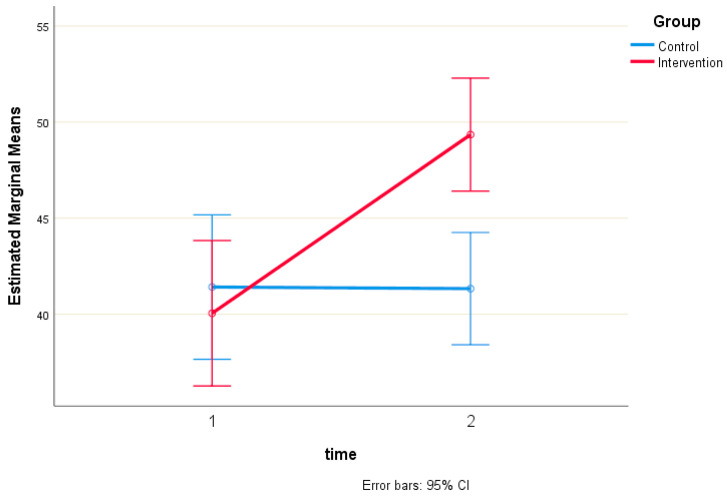
Estimated Marginal Means of PAM-13 score.

**Table 1 healthcare-13-00861-t001:** Characteristics of sample.

		GROUP					
		Control		Intervention		Total	
		Count	n %	Count	n %	Count	n %
Biological sex	Female	63	40.1%	71	45.8%	134	42.9%
	Male	94	59.9%	84	54.2%	178	57.1%
Education level	Primary	113	72.0%	115	74.2%	228	73.1%
	Secondary	20	12.7%	24	15.5%	44	14.1%
	Tertiary	24	15.3%	16	10.3%	40	12.8%
Comorbidity	No	13	8.3%	12	7.7%	25	8.0%
	Yes	144	91.7%	143	92.3%	287	92.0%

**Table 2 healthcare-13-00861-t002:** Short description of this study’s nurse-led intervention.

Intervention-Parts	Short Description	Methods/Tools	Expected Outcomes
Training Session	A 15 min session conducted by the first researcher, covering the disease, its treatment, and long-term management.	Researcher-led informational presentation and active participant discussion.	Enhanced patient understanding of the disease and treatment options.
Online Videos	Appropriately selected YouTube videos focusing on disease education, treatment, and self-management techniques (with Greek subtitles).	Videos from the World Glaucoma Association, including: “Understanding Glaucoma; World Glaucoma Week 2022; and “Patient Education Movie—World Glaucoma Week 2021“	Increased patient understanding of disease management through visual content.
Printed Material (Brochure)	Provision of a brochure containing essential information about glaucoma, treatment options, and self-management tips.	Printed brochure with text and graphics for patient education.	Improved patient knowledge regarding glaucoma and self-management strategies.

**Table 3 healthcare-13-00861-t003:** Measures of PAM-13 and HLS-EU16 scales (Mean ± Standard Deviation).

	Group		*p*-Value
	Control	Intervention	
PAM-13 score (pre)	41.4 ± 24.2	40.1 ± 23.7	*p* < 0.001
PAM-13 score (post)	41.3 ± 24.1	49.3 ± 10. 4	
HLS-EU16 score (pre)	10.3 ± 3.8	11.1 ± 3.5	*p* < 0.001
HLS-EU16 score (post)	10.5 ± 3.6	12.8 ± 2.6	

**Table 4 healthcare-13-00861-t004:** Multiple linear regression analysis of pre-intervention PAM-13 score (coefficients).

Model	Unstandardized Coefficients		Standardized Coefficients	t	Sig.	95.0% Confidence Interval for B		Collinearity Statistics	
	B	Std. Error	Beta			Lower Bound	Upper Bound	Tolerance	VIF
(Constant)	−2.113	8.847		−0.239	0.811	−19.521	15.296		
HLS-EU16 score (pre)	3.307	0.318	0.510	10.391	0.000	2.681	3.933	0.974	1.026
Group	−3.772	2.337	−0.079	−1.614	0.108	−8.370	0.826	0.981	1.019
Biological sex	−0.900	2.351	−0.019	−0.383	0.702	−5.526	3.726	0.989	1.011
Age	0.104	0.090	0.062	1.156	0.249	−0.073	0.281	0.804	1.243
Education level	3.863	1.844	0.114	2.095	0.037	0.235	7.492	0.794	1.259
Comorbidity	−2.495	4.271	−0.028	−0.584	0.560	−10.900	5.910	0.996	1.004

Dependent Variable: PAM-13 score (pre-intervention).

**Table 5 healthcare-13-00861-t005:** Multiple linear regression analysis of post-intervention PAM-13 score (coefficients).

Model	Unstandardized Coefficients		Standardized Coefficients	t	Sig.	95.0% Confidence Interval for B		Collinearity Statistics	
	B	Std. Error	Beta			Lower Bound	Upper Bound	Tolerance	VIF
(Constant)	11.868	8.453		1.404	0.161	−4.765	28.501		
HLS-EU16 score (post)	1.261	0.330	0.222	3.818	0.000	0.611	1.912	0.870	1.150
Group	5.356	2.198	0.141	2.437	0.015	1.031	9.682	0.879	1.137
Biological sex	1.187	2.100	0.031	0.565	0.572	−2.946	5.320	0.983	1.018
Age	0.093	0.080	0.070	1.165	0.245	−0.064	0.250	0.810	1.235
Education level	2.736	1.636	0.102	1.673	0.095	−0.482	5.955	0.801	1.249
Comorbidity	0.312	3.811	0.004	0.082	0.935	−7.187	7.810	0.992	1.008

Dependent Variable: PAM-13 score (post-intervention).

## Data Availability

The data that support the findings of this study are available from the corresponding author upon reasonable request due to privacy restrictions.

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
