# Peer review of "Enhancing Health Literacy and Self-Management in Glaucoma Patients: Evidence from a Nurse-Led Educational Intervention"

_healthcare, 2025, doi:10.3390/healthcare13080861_

Round 1
Reviewer 1 Report
Comments and Suggestions for Authors
To the authors,
The abstract could be improved by explicitly stating the study design at the beginning, which would strengthen methodological understanding for more technically inclined readers.
The methodology requires clarification in several areas. The inclusion criteria for the sample are not entirely clear. Although the diagnostic criteria are described, the process for verifying these criteria for sample inclusion is not fully explained. There are lingering doubts regarding how the diagnosis was confirmed.
The article has some limitations. The use of convenience sampling limits the generalisability of the findings. A formal randomisation process would have provided a more robust methodological approach.
Fundamentally, the main focus of the article appears to be therapeutic adherence, yet this is only marginally addressed. It should be more clearly included and contextualised in the introduction or background section of the paper.
Author Response
Dear Reviewer 1,
Thank you very much for your valuable comments, which have significantly contributed to the improvement of our paper. Please find attached our point-by-point responses to your suggestions. We believe that all the issues you raised have now been fully addressed, and the revisions meet your expectations.
On behalf of all authors,
Dr. Konstantinos Giakoumidakis
Associate Professor, Department of Nursing
School of Health Sciences, Hellenic Mediterranean University, Heraklion, Crete, Greece

Reviewer 2 Report
Comments and Suggestions for Authors
Congratulations on a nicely conducted study with a valuable message for eye health care workers who treat glaucoma.
Lines 204-205: The text states that the PAM has been tested and verified for a Greek population. In the section immediately before this, even though a Greek version of the HLS-16 EU was used in the study, this claim is not made. Has this questionnaire been validated for a Greek population? If so, it should be stated. If not, an explanation is needed to justify its use nonetheless.
Author Response
Dear Reviewer 2,
Thank you very much for your valuable comments, which have significantly contributed to the improvement of our paper. Please find attached our point-by-point responses to your suggestions. We believe that all the issues you raised have now been fully addressed, and the revisions meet your expectations.
On behalf of all authors,
Dr. Konstantinos Giakoumidakis
Associate Professor, Department of Nursing
School of Health Sciences, Hellenic Mediterranean University, Heraklion, Crete, Greece

Reviewer 3 Report
Comments and Suggestions for Authors
This study evaluates the effectiveness of a nurse-led educational intervention in enhancing health literacy and self-management among glaucoma patients in Greece. A randomized controlled design involving 312 patients was employed. The study found statistically significant improvements in both health literacy and patient activation levels post-intervention.
There are several concerns: 1. Study is confined to a single tertiary hospital in Crete. Encourage multicenter validation to enhance external validity. 2. All outcome measures are self-reported. Where feasible, complement with objective adherence measures (e.g., pharmacy refill rates). 3. Some graphs (e.g., Figures 1–2) are referenced but lack full clarity in labeling. Improve figure captions and ensure consistency with text. 4. Reflect more deeply on cultural factors affecting health literacy in Greek populations in discussion section.
Author Response
Dear Reviewer 3,
Thank you very much for your valuable comments, which have significantly contributed to the improvement of our paper. Please find attached our point-by-point responses to your suggestions. We believe that all the issues you raised have now been fully addressed, and the revisions meet your expectations.
On behalf of all authors,
Dr. Konstantinos Giakoumidakis
Associate Professor, Department of Nursing
School of Health Sciences, Hellenic Mediterranean University, Heraklion, Crete, Greece
